# Landfill Site Selection for Medical Waste Using an Integrated SWARA-WASPAS Framework Based on Spherical Fuzzy Set

**Saeid Jafarzadeh Ghoushchi** [1] , **Shabnam Rahnamay Bonab** [1] , **Ali Memarpour Ghiaci** [2] ,
**Gholamreza Haseli** [3] , **Hana Tomaskova** [4,*] **and Mostafa Hajiaghaei-Keshteli** [5]

1 Faculty of Industrial Engineering, Urmia University of Technology, Urmia 57166, Iran;
s.jafarzadeh@uut.ac.ir (S.J.G.); ShabnamRahnamay@ine.uut.ac.ir (S.R.B.)
2 Industrial Engineering Department, Malek Ashtar University of Technology, Tehran 15875, Iran;
alimemarpour@ine.uut.ac.ir
3 Department of Management, Faculty of Economic, Management and Social Science, Shiraz University,
Shiraz 71345, Iran; ghr.haseli@gmail.com
4 Faculty of Informatics and Management, University of Hradec Kralove,
50003 Hradec Kralove, Czech Republic
5 Tecnologico de Monterrey, Escuela de Ingeniería y Ciencias, Puebla 38115, Mexico; mostafahaji@tec.mx
* Correspondence: hana.tomaskova@uhk.cz

**Abstract:** Selecting suitable locations for the disposal of medical waste is a serious matter. This study aims to propose a novel approach to selecting the optimal landfill for medical waste using Multi-Criteria Decision-Making (MCDM) methods. For better considerations of the uncertainty in choosing the optimal landfill, the MCDM methods are extended by spherical fuzzy sets (SFS). The identified criteria affecting the selection of the optimal location for landfilling medical waste include three categories; environmental, economic, and social. Moreover, the weights of the 13 criteria were computed by Spherical Fuzzy Step-Wise Weight Assessment Ratio Analysis (SFSWARA). In the next step, the alternatives were analyzed and ranked using Spherical Fuzzy Weighted Aggregated Sum Product Assessment (SFWASPAS). Finally, in order to show the accuracy and validity of the results, the proposed approach was compared with the IF-SWARA-WASPAS method. Examination of the results showed that in the IF environment the ranking is not complete, and the results of the proposed method are more reliable. Furthermore, ten scenarios were created by changing the weight of the criteria, and the results were compared with the proposed method. The overall results were similar to the SF-SWARA-WASPAS method.

**Keywords:** MCDM; medical waste; SWARA; WASPAS; spherical fuzzy sets

## 1. Introduction

Lately, with the growth of population, waste generation in various types has increased a lot, all because of the actions of human beings, changes that are related to their patterns of utilization [1]. The quick increase in world population, urbanization, high consumption of materials in developing countries, increasing product complexity, use of substances that may cause environmental problems in the production of consumer goods, lifestyle, income level, and people's behaviors require a waste management system [2]. Infectious wastes are wastes in which bacteria, viruses, parasites, or fungi are present in enough quantities to cause disease in susceptible hosts. Moreover, lesions contaminated with blood or body fluids, tissues, patient organs, and sharp objects utilized during treatment are considered medical waste [3]. Waste generated in hospitals, clinics, and health care settings or environments contaminated with patient secretions, bodily fluids, such as blood, sputum, saliva, and urine, are considered medical waste. Historically, concerns about the potential health risks of medical waste in the United States were first discussed in the 1980s [4]. Due to increased waste production and high transportation costs, especially in developing countries, waste management has become one of the main challenges in urban

planning [5,6]. Municipal waste management is one of the most serious problems in the world despite the development of technology. The last step in waste management is waste disposal, and the removal of waste from the human environment occurs by changing it into materials that are no longer waste. This stage is of great importance because by using appropriate disposal methods, it can prevent many problems, including the production of various pollutants and the resulting health and environmental issues [7]. Among waste disposal methods, landfilling has been the most necessary method of disposal in some countries and is still utilized in developed countries [8]. In particular, most developing countries, such as Cambodia, Vietnam, India, the Philippines, Bangladesh, Thailand, Malaysia, Indonesia and Palestine, dispose of solid waste in landfills with poor management [9]. This is another example in which improper management of contaminated personal protective equipment (PPE) and health waste may increase viral diseases in the environment [10]. As a result, one of the problems that inevitably arise if medical waste is not correctly managed is severe diseases and environmental problems [11].

Medical waste, if stored, can cause environmental problems and threaten the general health of humans, so sanitary waste must be disposed of. Until almost 50 or 60 years ago, the world did not pay attention to landfilling, and in most parts of the world, the same traditional methods were utilized to accumulate and collect garbage and burn it in open areas. The general idea of the people was that in this way they reduce the amount of waste and pollution caused by them. Gradually, experts in this field realized that in this way, the pollution is changed from one type to another because burning waste in the open air causes the pollution of these materials [12]. One of the best methods of disposal is burial. Choosing a suitable place for burying sanitary waste has a special process because it affects human health and environmental sustainability, and to choose a suitable place for burial, several factors, such as political and economic, must be considered. Identifying a suitable landfill for hazardous waste requires appropriate and standardized criteria to propose the necessary facilities from an environmental, social, and economic perspective [13]. Sener et al. [14] utilized criteria, such as geology, hydrology, land use, slope, and altitude. Chamchali and Ghazi Fard [15] also considered soil erosion as a criterion for landfilling. Therefore, the selection of a landfill can be assumed as a Multi-Criteria Decision-Making (MCDM) issue. Decision-making can be considered as the optimal choice among alternatives according to a number of criteria [16,17]. In the recent past, MCDM methods have been studied. For example, Geographical Information Systems (GIS) and MCDM methods have been used to assess locations of solar farms, and a case study has been conducted in southeastern Spain [18]. Mavi et al. [19] have utilized the integrated The Wise Weight Assessment Ratio Analysis (SWARA) and Multi-Objective Optimization on the basis of the Ratio Analysis (MOORA) approach in a fuzzy environment to select a third-party reverse logistics provider in the plastics industry and the Technique for Order of Preference by Similarity to Ideal Solution (TOPSIS) method has been utilized to select the location of wind farms in the intuitive fuzzy environment in Turkey [20]. Haseli et al. [21] have proposed a base-criterion method (BCM) to solve multi-criteria decision-making problems. Jafarzadeh Ghoushchi et al. [22] have used MCDM methods for sustainable supplier selection in the oilseed industry. Rani et al. [23] have utilized SWARA and VIKOR methods for the performance evaluation of solar panels in the Pythagorean Fuzzy (PF) environment. Fuzzy environments have also been used to classify and retrieve information [24]. Wang et al. have used the TOPSIS method in fuzzy environments to evaluate the renewable energy production capabilities [25]. The Spherical Fuzzy Set (SFS) was introduced by Gündoğdu and Kahraman [26]. It is a three-dimensional fuzzy set introduced as an extension of the Pythagorean fuzzy set (PFS), Intuitionistic fuzzy sets (IFS), and Neutrosophic Sets (NS), especially to control the uncertainty during the quantification of expert quantification. In IFS, the sum of the membership and non-membership degrees must be between 0 and 1, and the hesitance degree is obtained by subtracting this value from 1 [27]. These sets calculate the uncertainty degree more accurately than type-1 and type-2 fuzzy sets. In PFS, the sum of membership and non-membership degrees is allowed to be larger than

1; however, the sum of the squares of the membership and non-membership degrees must be equal to or less than 1 [28]. In SFS, the sum of the membership, non-membership, and indeterminacy/neutral degrees can be larger than 1. However, the sum of squares of the membership, non-membership, and neutral degrees must be between 0 and 1, which makes SFS nonlinear. Moreover, the membership, non-membership, and hesitance degrees are specified individually, which permits decision-makers (DMs) to express the decision-making issue using more information about the criteria (higher flexibility) contrasted to IFS and PFS. SFS makes the decision-making procedure smarter (equivalent to human judgment), such that using SFS can lead to a more accurate alternative evaluation in the decision-making process [29]. The major difference between PFS and SFS is that in SFS, we study a neutral degree, whereas in PFS, we do not [30]. SFSs were lately proposed as a step to model problems more accurately on the basis of human nature, thus expanding the space of membership levels are defined. Because of this benefit, SFS has newly been utilized in a number of applications, such as storage location selection [31], waste disposal location selection [32], and design evaluation and technology of a linear delta robot [33]. Ashraf et al. [34] have provided spherical fuzzy Dombi aggregation operators and their application in group decision-making problems, and has used spherical fuzzy to detect COVID-19 [35].

In this research, we expand the combination of two widely utilized methods among MCDM methods, SWARA and The Weighted Aggregated Sum Product Assessment (WASPAS), in spherical fuzzy environments and show their application through a problem of choosing the optimal location for the disposal of medical waste. The reason for using SFS is to allow DMs to specify a membership function on a spherical area in order to generalize other fuzzy set components and assign the membership performance parameters independently with a larger domain. Therefore, the developed approach can purpose more reasonable and accurate results using the advantages of the SFS set, which reflects uncertainty in a more appropriate way and is equivalent to the judgment of decision-makers. The membership function degrees of the spherical fuzzy set can fully express individuals' decision-making awareness and accurately describe the decision information with a parameter that can flexibly regulate the range of information expression [36]. If medical waste is not buried in the right site, it will adversely affect the surface water, groundwater, and air and soil. The weight of the criteria is one of the important concerns in decision-making issues, and in this study, considering the environmental issues, the opinions of DMs are very important. For this reason, the SWARA method is used, which is an efficient method, and with the knowledge provided by DMs, the weight of the criteria is obtained. The main advantage of this method is estimating the accuracy of experts in determining the weight of the criteria. The WASPAS method, which is a combination of two unique techniques in MCDM, has been used to rank landfills for sanitary waste. The accuracy of this method in comparison with other independent methods is noteworthy. Our goal in this study is to investigate and assess medical waste landfills in order to prevent environmental pollution by choosing a suitable location and do not affect the sustainability of the environment, and not threaten human health.

The rest of this study is as follows: In Section 2, a literature review is provided. In Section 3, the introductions, including the concept of spherical fuzzy set and SFSWARA weighting method and SFWASPAS ranking method, are introduced. In Section 4, the proposed method is presented. In Section 5, a case study is introduced, the proposed method is implemented, and the analysis of the results is presented. Eventually, in Section 6, conclusions and suggestions for the development of this study are presented.

## 2. Literature Review

### 2.1. SWARA Method

The Wise Weight Assessment Ratio Analysis (SWARA) method has been proposed by Keršuliene et al. [37]. In this method, the most important criterion is ranked first and the least important criterion is given the last rank. In this method, experts (respondents) have

the main role in determining the weight of the criteria. The main feature of this method is the possibility of estimating experts and pundits in relation to the importance of the criteria in the process of determining their weight [37]. The most important benefit of this method in decision-making is that in some problems, priorities are defined based on the policies of companies or countries and do not need to be evaluated to rank criteria. SWARA gives decision-makers and policymakers the opportunity to prioritize based on the current state of the environment and the economy. In other methods, such as the Analytic Hierarchy Process (AHP) or Analytic Network Process (ANP), our model is based on expert criteria and evaluations that affect rankings. Therefore, SWARA can be helpful for some cases where the priorities are in accordance with the known situations. Finally, SWARA has been proposed to be utilized in various decision environments [38]. Researchers have utilized this method in recent years in various fields. The SWARA method has been used to assess agile supplier selection criteria [39] and has been utilized to design bottle packaging to improve sales [40]. A combination of SWARA and Occupational Repetitive Actions (OCRA) methods has been utilized to rank hotels [41]. Karabašević et al. [42] have utilized SWARA and ARAS methods for personnel selection. Maghsoodi [43] has utilized the SWARA method to select the optimal renewable energy technology. SWARA and GRA methods have also been used to prioritize the failures in Solar Panel Systems under Z-Information [44]. In this method, experts have an important role in determining the weight of criteria and is useful for collecting information and coordinating information obtained from experts. It is a simple method that has been selected for weighting criteria in this study.

Due to various factors, for example, lack of complete information, qualitative judgment of experts, and to deal with uncertainty, decisions have been made in a fuzzy environment [45]. Mavi et al. [19] developed Fuzzy SWARA and included uncertainty in this method. In his research, he utilized the SWARA method to weight the criteria and MULTIMOORA method under the fuzzy set to rank third-party reverse logistics providers in the plastics industry. Karabašević et al. [46] have proposed SWARA and ARAS methods for evaluating personnel in a fuzzy framework. Mishra [47] has utilized the integrated SWARA-WASPAS method to select a Bioenergy Production (BPT) under the intuitive fuzzy set. All of these benefits have led to the usage of this efficient method in the present study for weighting risk criteria. Based on the outcomes of this study, it can be said that spherical fuzzy SWARA is more efficient than the usual SWARA method, and uncertainty has decreased significantly.

### 2.2. WASPAS Method

The Weighted Aggregated Sum Product Assessment (WASPAS) is a well-known and efficient solution for solving problems, which was introduced by Zavadskas [48]. In fact, this method is a combination of two MCDM techniques, Weighted Sum Model (WSM) and Weighted Product Model (WPM), and due to its easiness and mathematical ability to propose accurate outcomes in contrast to WSM and WPM, it has been widely welcomed, and with a value of $\lambda$ as an interface, the ranking is performed based on two final indicators. Many researchers have utilized the WASPAS method to dissolve multi-criteria decision issues. Dėjus et al. [49] have utilized the classic WASPAS to solve the problem of job security. Classic WASPAS with a different $\lambda$ for decision-making, the selection of flexible manufacturing systems (FMS), the selection of a machine for flexible production systems, the selection of automated guided vehicles (AGV), the selection of an automatic inspection system, and the selection of robots has been utilized [50]. Some researchers have expanded WASPAS to fuzzy sets. To solve the problem of locating the solar wind power plant, the WASPAS method has been utilized in the Interval Neutrosophic (IN) environment [51]. To assess the performance of retail stores, the WASPAS method has been utilized in Pythagorean fuzzy environments, and the results have been compared with classic WASPAS and intuitionistic fuzzy WASPAS [52]. The SWARA method has been utilized to weight the criteria and WASPAS in Pythagorean fuzzy environments and has

been utilized to assess and prioritize sustainable suppliers in manufacturing companies [53]. F-BWM and F-WASPAS approaches have been employed to assess the risk factors of SOFC devices [54]. Fuzzy set theories are utilized for uncertainty in the decision-making process. This method has been selected as the ranking method in this study due to the combination of two unique MCDM methods and more accuracy than other independent methods, as well as due to the ease and mathematical ability to provide accurate results.

Due to various factors, such as lack of complete information, the qualitative judgment of specialists that cause uncertainty, and accuracy in decision-making, and because conventional MCDM methods in the environment of definite numbers cannot solve the problem with such vague information. Fuzzy multi-criteria methods have been developed to deal with uncertainty and assess the weights of criteria and rank options [55]. SFS are extensions of intuitive and PFS in order to create more freedom for specialists and reduce inaccuracies in the outcomes. Based on the outcomes of this study, it can be said that spherical fuzzy WASPAS (SFWASPAS) is more efficient than the usual method and uncertainty is significantly reduced. In this proposed approach to select the suitable site for landfilling medical waste, degree of membership, non-membership, and doubt of DM opinions are also considered independently, and the results are more reasonable and closer to the real world.

## 3. Preliminaries

The purpose of this study is to present a novel decision approach based on spherical fuzzy theory to select a suitable landfill for medical waste. Therefore, the theory used in this proposed approach is discussed in this section.

### 3.1. SFS

One of the latest fuzzy sets is the SFS, proposed by Gündoğdu and Kahraman [26]. SFSs are extensions of PFSs and provide specialists with a larger domain [33]. Some of the principles of SFSs and their operation are presented in this section.

**Definition 1.** *According to Ref. x, an SPS S is in the following form:*

$$S = [(x(\mu_s(x)v_s(x)\pi_s(x))) \; | x \in X] \tag{1}$$

In this relationship, $\mu_s : X \to [0,1]$ $v_s : X \to [0,1]$ $\pi_s : X \to [0,1]$, respectively, represent the degrees of membership, non-membership, and hesitance for every $x \in X$ in SPS $S$, and the following condition holds:

$$0 \le (\mu_s(x))^2 + (v_s(x))^2 + (\pi_s(x))^2 \le 1 \tag{2}$$

**Definition 2.** *Let $S1 = [\mu_{s1}v_{s1}\pi_{s1}]$ and $S2 = [\mu_{s2}v_{s2}\pi_{s2}]$ be two SF numbers and k be a constant number greater than zero. In this case, the mathematical operations of these two SF numbers are performed via the following equations.*

$$S1 \oplus S2 =$$
$$= \left[ \sqrt{\mu_{S1}^2 + \mu_{S2}^2 - \mu_{S1}^2\mu_{S2}^2})v_{s1}v_{s2} \sqrt{((1 - \mu_{S2}^2)\pi_{s1} + (1 - \mu_{S1}^2)\pi_{s2} - \pi_{s1})\pi_{s2}} \right] \tag{3}$$

$$S1 \otimes S2 =$$
$$= [\mu_{s1}\mu_{s2} \sqrt{(v_{s1}^2 + v_{s2}^2 - v_{s1}^2 v_{s2}^2)} \sqrt{((1 - v_{s2}^2)\pi_{s1}^2 + (1 - v_{s1}^2)\pi_{s2}^2 - \pi_{s1}^2 \pi_{s2}^2)}] \tag{4}$$

$$kS = [\sqrt{(1 - (1 - \mu_s^2)^k)}v_s^2 \sqrt{((1 - \mu_s^2)^k - (1 - \mu_s^2 - \pi_s^2)^k)}] \tag{5}$$

$$S^k = \mu_s^k \sqrt{(1 - (1 - v_s^2)^k)} \sqrt{((1 - v_s^2)^k - (1 - v_s^2 - \pi_s^2)^k)} \tag{6}$$

**Definition 3.** *Let $S1 = [\mu_s1 v_s1 \pi_s1]$ and $S2 = [\mu_s2 v_s2 \pi_s2]$ be two SF numbers. The following rules with the condition $k_1 k_2 > 0$. k hold for SF numbers.*

$$S1 \oplus S2 = S2 \oplus S1 \tag{7}$$

$$S1 \otimes S2 = S2 \otimes S1 \tag{8}$$

$$k(S1 \oplus S2) = kS1 \oplus kS2 \tag{9}$$

$$k_1 S1 + k_2 S1 = (k_1 + k_2)S1 \tag{10}$$

$$(S1 \otimes S2)^k = S1^k \otimes S2^k \tag{11}$$

$$S1^{k_1} \otimes S1^{k_2} = S1^{(k_1 + k_2)} \tag{12}$$

**Definition 4.** *Let $S = \mu_s v_s \pi_s$ represent an SF number. The score value and accuracy function of the number S are computed as follows:*

$$Score(S) = (\mu_S - \pi_S)^2 - (v_S - \pi_S)^2 \tag{13}$$

$$Accuracy(S) = \mu_s^2 + v_s^2 + \pi_s^2 \tag{14}$$

Note that: $S1 < S2$ if and only if:

$$\begin{aligned} &i. score(S1) < score(S2) \\ &ii. score(S1) = score(S2) \, and \, Accuracy(S1) < Accuracy(S2) \end{aligned} \tag{15}$$

**Definition 5.** *Given $w = (w_1, w_2, s, w_n) w_i \in [0,1]$ ; $\sum_{i=1}^{n} w_i = 1$, the spherical weighted arithmetic mean (SWAM) is computed as follows:*

$$SWAM_w(S1, \cdots, Sn) = w_1 S1 + w_2 S2 + \cdots + w_n Sn =$$
$$[1 - \Pi_{i=1}^n (1 - \mu_s^2)^{wi}]^{1/2} \Pi_{i=1}^n v_s^{wi} [\Pi_{i=1}^n (1 - \mu_s^2)^{wi} - \Pi_{i=1}^n (1 - \mu_s^2 - \pi_s^2)^{wi}]^{1/2} \tag{16}$$

**Definition 6.** *Given $w = (w_1, w_2, \cdots, w_n) w_i \in [0,1]; \sum_{i=1}^{n} w_i = 1$, the spherical weighted geometric mean (SWGM) is computed as follows:*

$$SWGM_w(S1, \cdots, Sn) = S1^{w1} + S2^{w2} + \cdots + Sn^{wn} =$$
$$\Pi_{i=1}^n \mu_s^{wi} [1 - \Pi_{i=1}^n (1 - v_s^2)^{wi}]^{1/2} [\Pi_{i=1}^n (1 - v_s^2)^{wi} - \Pi_{i=1}^n (1 - v_s^2 - \pi_s^2)^{wi}]^{1/2} \tag{17}$$

*3.2. SF-SWARA*

The Gradual Weight Assessment Ratio (SWARA) analysis method has been proposed by Keršuliene et al. [37]. Various criteria, such as lack of complete information, the qualitative judgment of specialists, inaccessible information, and uncertainty, make the decision difficult, and conventional MCDM methods cannot be effective in solving problems, so the decision is made in a fuzzy environment [43]. The aim of this article is to expand the SWARA method to SFSWARA, which is a more powerful approach to problem-solving. A brief description of the SFSWARA steps is provided below.

Step 1. Determining the appropriate criteria by specialists.

At this step, the specialists detect and assess the appropriate criteria. The decision criteria are expressed as a set of $T = T_1, T_2, \cdots, T_n$.

Step 2. Making a decision matrix with PFNs.

At this step, the linguistic variables expressed by the DMs are changed to Spherical Fuzzy numbers with Table 1.

**Table 1.** Linguistic terms and their corresponding spherical fuzzy numbers.

| Linguistic Terms | Spherical Fuzzy Number | | |
| --- | --- | --- | --- |
| | $\mu$ | v | $\pi$ |
| Absolutely More Importance (AMI) | 0.9 | 0.1 | 0.1 |
| Very High importance (VHI) | 0.8 | 0.2 | 0.2 |
| High Importance (HI) | 0.7 | 0.3 | 0.3 |
| Slightly more importance (SMI) | 0.6 | 0.4 | 0.4 |
| Equally Importance (EI) | 0.5 | 0.5 | 0.5 |
| Slightly Low Importance (SLI) | 0.4 | 0.6 | 0.4 |
| low Importance (LI) | 0.3 | 0.7 | 0.3 |
| Very Low Importance (VLI) | 0.2 | 0.8 | 0.2 |
| Absolutely Low Importance (ALI) | 0.1 | 0.9 | 0.1 |

Step 3. Calculating the weight of DMs and make the aggregated Decision matrix. The preferences of each DM are aggregated using the SWAM or SWGM operator, as shown in Equations (16) and (17).

Step 4. Calculation of SCORE VALUES ($S_j$) and sorting criteria.

To sort the criteria, the SCORE VALUE is computed by Equation (13). The criteria of interest are written in order based on the SCORE VALUE. The most and the least important criteria are placed in higher and lower categories, respectively.

Step 5. Determining the relative importance of criteria ($S_j$) and calculating the coefficient ($K_j$).

In this step, the relative importance of each criterion is specified compared to previous criteria. The coefficient $K_j$ is a function of the relative importance of each criterion, which is computed using the following equation:

$$K_j = \begin{cases} 1 & j = 1 \\ s_j + 1 & j > 1. \end{cases} \tag{18}$$

Step 6. Calculating the primary weight of each criterion.

The primary weight of each criterion is computed by the following equation, where it should be borne in mind that the first criterion's weight, which is the most important criterion, is considered equal to 1.

$$q_j = \begin{cases} 1 & j = 1 \\ \frac{k_{j-1}}{k_j} & j > 1. \end{cases} \tag{19}$$

Step 7. Calculating the final weight of criteria.

In the last step, the final weight of the criteria is computed through the following equation, and it is also called the nominal weight.

$$\omega_j = \frac{q_j}{\sum_{j=1}^{n} q_j} \tag{20}$$

### 3.3. SF-WASPAS

Step 1: Construction of the decision matrix.

Let us assume that $X = x_1, x_2, \cdots, x_i, \cdots, x_m$ and $C = C_1, C_2, \cdots, C_j, \cdots, C_n$, which imply the set of alternatives and set of criteria, respectively, and that a set of $DMs D = D_1, D_2, \cdots, D_K$ has been formed. Let $R = (x_{ij}^k)$ $i = 1(1)m$ $j = 1(1)n$ be decision matrix expressed by the DMs. Hence, $x_{ij}^k$ is the evaluation of alternative $X_i$ about criterion $C_j$ by the $k^{th}$ specialist. For an MCDM problem, the decision matrix based on spherical fuzzy must be constructed as in Equation (21).

$$R_{ij} = (C_j(x_i))_{m \times n} = \begin{pmatrix} (\mu_{11}v_{11}\pi_{11}) & \cdots & (\mu_{1n}v_{1n}\pi_{1n}) \\ \vdots & \ddots & \vdots \\ (\mu_{m1}v_{m1}\pi_{m1}) & \cdots & (\mu_{mn}v_{mn}\pi_{mn}) \end{pmatrix} \qquad (21)$$

Step 2: Conversion of linguistic variables to SF numbers.

In this step, the linguistic variables of the decision matrix are transformed to SF numbers considering the decision matrix in the first stage, and the decision matrix is computed using SF numbers. The conversion procedure is according to Table 1.

Step 3: Construction of an aggregated Decision matrix.

The preferences of each DM are aggregated using the SWAM or SWGM operator, as shown in Equations (16) and (17).

Step 4: Construction of the Weighted Sum Model (WSM) decision matrix.

The decision matrix for WSM, using Equation (22), is computed.

$$\widetilde{Q}_i^1 = \sum_{j=1}^{n} \widetilde{x}_{ijw} = \sum_{j=1}^{n} \widetilde{x}_{ij}\widetilde{w}_j \qquad (22)$$

Equation (22) can be separated into two parts for ease of operation. First, the multiplication operator and then the addition operator. Perform the multiplication part of Equation (22) using Equation (23).

$$\widetilde{x}_{ijw} = \widetilde{x}_{ij}\widetilde{w}_j =$$
$$= \left\langle \left(1 - (1 - \mu_{\widetilde{x}_{ij}}^2)^{\widetilde{w}_j}\right)^{1/2} v_{\widetilde{x}_{ij}}^{\widetilde{w}_j} \left((1 - \mu_{\widetilde{x}_{ij}}^2)^{\widetilde{w}_j} - (1 - \mu_{\widetilde{x}_{ij}}^2 - \pi_{\widetilde{x}_{ij}}^2)^{\widetilde{w}_j}\right)^{1/2} \right\rangle \qquad (23)$$

Perform the sum of Equation (22) using Equation (24).

$$\widetilde{x}_{i1w} \oplus \widetilde{x}_{i2w} = \left\langle (\mu_{\widetilde{x}_{i1w}}^2 + \mu_{\widetilde{x}_{i2w}}^2 - \mu_{\widetilde{x}_{i1w}}^2 \mu_{\widetilde{x}_{i2w}}^2)^{\frac{1}{2}} v_{\widetilde{x}_{i1w}}^2 v_{\widetilde{x}_{i2w}}^2 \right.$$
$$\left. \left((1 - \mu_{\widetilde{x}_{i2w}}^2)\pi_{\widetilde{x}_{i1w}}^2 + (1 - \mu_{\widetilde{x}_{i1w}}^2)\pi_{\widetilde{x}_{i2w}}^2 - \pi_{\widetilde{x}_{i1w}}^2 \pi_{\widetilde{x}_{i2w}}^2\right)^{\frac{1}{2}} \right\rangle \qquad (24)$$

Step 5: Construction of the Weighted Product Model (WPM) decision matrix.

$$\widetilde{Q}_i^{(2)} = \Pi_{j=1}^{n} \widetilde{x}_{ij}^{\widetilde{w}_j} \qquad (25)$$

Equation (25) can be separated into two parts for ease of operation. First, the exponential operator (power) and then the multiplication operator. Do the exponential part of Equation (25) using Equation (26).

$$x_{ij}^{\widetilde{w}_j} = \left\langle \mu_{\widetilde{x}_{ij}}^{\widetilde{w}_j} \left((1 - v_{\widetilde{x}_{ij}}^2)^{\widetilde{w}_j}\right)^{\frac{1}{2}} \left((1 - v_{\widetilde{x}_{ij}}^2)^{\widetilde{w}_j} - (1 - v_{\widetilde{x}_{ij}}^2 - \pi_{\widetilde{x}_{ij}}^2)^{\widetilde{w}_j}\right)^{\frac{1}{2}} \right\rangle \qquad (26)$$

Perform the multiplication part of Equation (25) using Equation (27).

$$\widetilde{x}_{i1w} \otimes \widetilde{x}_{i2w} = \left\langle \mu_{\widetilde{x}_{i1}}^{\widetilde{w}_1} \mu_{\widetilde{x}_{i1}}^{\widetilde{w}_2} \left(v_{\widetilde{x}_{i1}}^{\widetilde{w}_1^2} + v_{\widetilde{x}_{i2}}^{\widetilde{w}_2^2} - v_{\widetilde{x}_{i1}}^{\widetilde{w}_1^2} v_{\widetilde{x}_{i2}}^{\widetilde{w}_2^2}\right)^{\frac{1}{2}} \right.$$
$$\left. \left((1 - v_{\widetilde{x}_{i2}}^{\widetilde{w}_2^2})\pi_{\widetilde{x}_{i1}}^{\widetilde{w}_1^2} + (1 - v_{\widetilde{x}_{i1}}^{\widetilde{w}_1^2})\pi_{\widetilde{x}_{i2}}^{\widetilde{w}_2^2} - \pi_{\widetilde{x}_{i1}}^{\widetilde{w}_1^2} \pi_{\widetilde{x}_{i2}}^{\widetilde{w}_2^2}\right)^{\frac{1}{2}} \right\rangle \qquad (27)$$

Step 6: Specify the value of $\lambda$ and compute Equations (28) and (29):

$$\lambda\widetilde{Q}_i^{(1)} = \left\langle \left(1 - (1 - \mu_{\widetilde{Q}_i^{(1)}}^2)^{\lambda}\right)^{\frac{1}{2}} v_{\widetilde{Q}_i^{(1)}}^{\lambda} \left((1 - \mu_{\widetilde{Q}_i^{(1)}}^2)^{\lambda} - (1 - \mu_{\widetilde{Q}_i^{(1)}}^2 - \pi_{\widetilde{Q}_i^{(1)}}^2)^{\lambda}\right)^{\frac{1}{2}} \right\rangle \qquad (28)$$

$$(1-\lambda)\widetilde{Q}_i^{(2)} = \left\langle \left( 1 - (1-\mu_{\widetilde{Q}_i^{(2)}}^2)^{1-\lambda} \right)^{\frac{1}{2}} v_{\widetilde{Q}_i^{(1)}}^{1-\lambda} \right.$$
$$\left. \left( (1-\mu_{\widetilde{Q}_i^{(2)}}^2)^{1-\lambda} - (1-\mu_{\widetilde{Q}_i^{(2)}}^2 - \pi_{\widetilde{Q}_i^{(2)}}^2)^{1-\lambda} \right)^{\frac{1}{2}} \right\rangle \tag{29}$$

Step 7: Compute the value of $\widetilde{Q}_i$ for the $i$th option to rank the optimal landfill for medical waste.

$$\widetilde{Q}_i = \lambda \widetilde{Q}_i^{(1)} + (1-\lambda)\widetilde{Q}_i^{(2)} \tag{30}$$

Finally, the score values of $\widetilde{Q}_i$ are computed using Equation (13). Alternative options are ranked in descending order.

## 4. Proposed Approach

In this section, a novel approach to selecting a suitable location for the disposal of medical waste using MCDM methods in a spherical fuzzy environment is presented. According to the complete explanations of SF-SWARA and SFWASPAS that were presented in the previous section, the proposed approach is presented in two phases. In the first phase, according to the field and geographical situations of the region and former polls and specialist opinions, the selected criteria have been specified by using the SF-SWARA method; the criteria weight were specified by specialists and specialists. In the second phase, according to the outputs of the previous phase, the proposed sites were ranked by the SF-WASPAS method, and among the alternatives, the optimal sites for landfilling of medical waste were identified. The algorithm for solving the problem of locating medical waste landfills is shown in Figure 1.

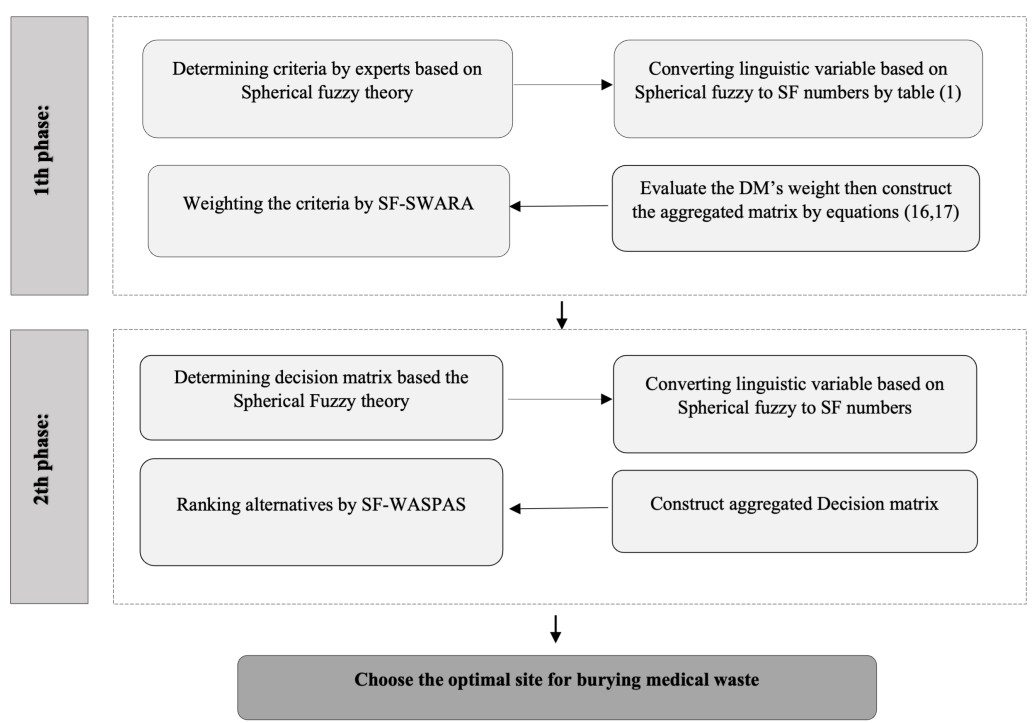

**Figure 1.** Flowchart of the proposed methodology.

## 5. Analyzing the Results

This section proposes descriptions for the case study, and the proposed method is implemented step by step based on the formulas presented in the method section.

### 5.1. Case Study

In order to investigate the feasibility of the proposed approach in this research to solve the location problem, it has been utilized to select the optimal landfill for medical waste

in a real area. Urmia is a big city in northwestern Iran, the capital of the province of West Azerbaijan (Figure 2, Point A in the map shows Urmia). According to the 2016 census, with a population of 736,224, it is the tenth most populous city in Iran. Urmia, with an altitude of 1332 m, is located west of Urmia Lake, at the foot of Garlic Mountain and in the middle of Urmia plain. The weather in Urmia is relatively hot in summer and cold in winter. Urmia, with its privileged geographical position, is located 20 km from Lake Urmia. It is surrounded by martyrs, Moon Mountain, Ali Panjeh Si Mountain, and Ali Iman Mountain. In fact, Urmia is located between Lake Urmia and the wall of the mountains in the west of the province. The city of Urmia is located on an orbit of 37 degrees and 32 min in the northern hemisphere of the equator. It is also located on a 45-degree meridian 2 min east of the Greenwich meridian. In the city of Urmia, about 2.5 to 3 tons of infectious waste is produced daily, and all private and public hospitals in Urmia are equipped with a device for decontaminating infectious waste, and infectious waste is transferred to the landfill after decontamination. Hospital wastes include infectious wastes, pathological wastes, sharp objects, pharmaceutical wastes, carcinogenic wastes, chemical wastes, radioactive wastes, high-pressure gas capsules, and wastes containing heavy metals.

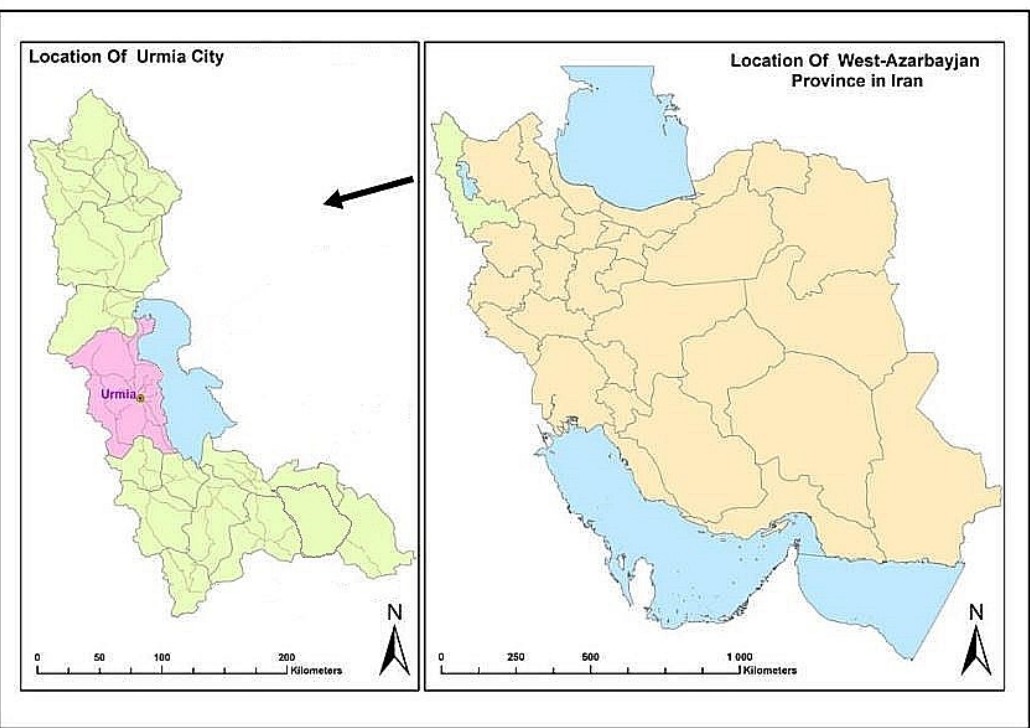

**Figure 2.** Geographical location of Urmia in the map of Iran.

Based on studies related to the city of Urmia and the field and geographical situations of the region and previous surveys and expert views, as well as the environmental standards of the Environment Organization of Iran (IDOE), a set of thirteen criteria (Distance from surface waters (C1), Groundwater depth (C2), Distance from the fault (C3), Geomorphology type (C4), Temperature (C5), Slope of the area (C6), Rainfall (C7), Material of bedrock (C8), Land use (C9), Distance from towns and villages (C10), Social acceptance (C11), Health risk (C12), and Noise (C13)) have been utilized to select an appropriate landfill for medical waste. The introduced criteria are based on three categories:

(A) Environmental criteria, C1 to C8,
(B) Economic criteria, C9 and C10, and
(C) Social criteria, C11 to C13.

Sanitary landfilling is an effective and proven method of disposing of sanitary waste and should be done in a way that does not endanger the environment. Therefore, special-

ized and experienced people in the field of environmental protection, agriculture, natural resources, and municipality have been used.

*5.2. Results*

In this section, the outcomes of the suggested research approach in selecting the optimal suitable landfill for medical waste are reviewed. The important purpose of this study is to introduce a novel solution for selecting a suitable site for landfilling medical waste using MCDM methods in an environment of uncertainty. In the first step, in order to speed up the decision-making process and prevent receiving additional data, it is essential to compute the appropriate and inappropriate distance for each of the criteria. For this purpose, the importance of each criterion has been collected through a questionnaire by specialists and experts in this field. Based on the SFSWARA method, the value of each criterion is expressed by relevant experts in the form of verbal variables using Table 1, which is shown in Table 2.

**Table 2.** The importance weights of the criteria.

| Criteria | DM1 | DM2 | DM3 |
|---|---|---|---|
| Distance from surface waters (C1) | HI | VHI | HI |
| Groundwater depth (C2) | SLI | LI | SLI |
| Distance from the fault (C3) | LI | VLI | VLI |
| Geomorphology type (C4) | HI | SMI | SMI |
| Temperature (C5) | ALI | ALI | ALI |
| Slope of the area (C6) | SMI | SMI | EI |
| Rainfall (C7) | ALI | VLI | VLI |
| Material of bedrock (C8) | LI | LI | VLI |
| Land use (C9) | LI | VLI | LI |
| Distance from towns and village (C10) | VLI | VLI | VLI |
| Social acceptance (C11) | SLI | LI | LI |
| Health risk (C12) | LI | SLI | LI |
| Noise (C13) | VLI | LI | VLI |

Then, they are transformed to spherical fuzzy numbers, and the opinions of decision-makers are merged using SWAM or SWGM operators considering the weight of the DMs, shown in Table 3.

**Table 3.** Aggregation of criteria weights based on SWAM operator.

| Criteria | Weights of Each Criterion | | |
|---|---|---|---|
| | $\mu$ | v | $\pi$ |
| C1 | 0.724098 | 0.276632 | 0.301999 |
| C2 | 0.382681 | 0.618786 | 0.400123 |
| C3 | 0.235121 | 0.768586 | 0.256541 |
| C4 | 0.634148 | 0.366926 | 0.352104 |
| C5 | 0.100000 | 0.900000 | 0.100000 |
| C6 | 0.554238 | 0.447214 | 0.450469 |
| C7 | 0.176343 | 0.828773 | 0.158943 |
| C8 | 0.255606 | 0.748331 | 0.257106 |
| C9 | 0.283219 | 0.718946 | 0.300038 |
| C10 | 0.200000 | 0.800000 | 0.200000 |
| C11 | 0.334053 | 0.668365 | 0.356238 |
| C12 | 0.32319 | 0.678748 | 0.300087 |
| C13 | 0.224087 | 0.778918 | 0.200024 |

Then, the SCORE VALUE is computed by Equation (13). The method of obtaining the weight of the criteria through the SFSWARA method and the final weight of the 13 criteria are presented in Table 4.

**Table 4.** The weights of the thirteen criteria

| Criteria | Score Value | $s_j$ | $k_j$ | $p_j$ | $w_j$ |
|----------|-------------|-------|-------|-------|-------|
| C1 | 0.177524 | - | 1 | 1 | 0.107464 |
| C4 | 0.079329 | 0.098195 | 1.098195 | 0.910585 | 0.097855 |
| C6 | 0.010757 | 0.068572 | 1.068572 | 0.852151 | 0.091576 |
| C2 | −0.04751 | 0.058267 | 1.058267 | 0.805233 | 0.086534 |
| C11 | −0.09693 | 0.049422 | 1.049422 | 0.767311 | 0.082459 |
| C12 | −0.14285 | 0.045919 | 1.045919 | 0.733624 | 0.078838 |
| C9 | −0.1752 | 0.032351 | 1.032351 | 0.710634 | 0.076368 |
| C8 | −0.2413 | 0.066098 | 1.066098 | 0.666574 | 0.071633 |
| C3 | −0.26173 | 0.020431 | 1.020431 | 0.653228 | 0.070199 |
| C13 | −0.33454 | 0.072807 | 1.072807 | 0.608896 | 0.065435 |
| C10 | −0.36 | 0.025461 | 1.025461 | 0.593778 | 0.06381 |
| C7 | −0.44837 | 0.088369 | 1.088369 | 0.545566 | 0.058629 |
| C5 | −0.64 | 0.191631 | 1.191631 | 0.457832 | 0.049201 |

In the second step, to select the optimal site for burying medical wastes, according to the SF-WASPAS method, the first decision matrix in the form of verbal variables is formed by the relevant specialists (Table 5).

**Table 5.** The decision matrix in the form of spherical fuzzy.

| | DM1 | DM2 | DM3 | DM1 | DM2 | DM3 | DM1 | DM2 | DM3 |
|---|-----|-----|-----|-----|-----|-----|-----|-----|-----|
| | | C1 | | | C2 | | | C3 | |
| L1 | HI | HI | HI | HI | VHI | VHI | SLI | EI | SLI |
| L2 | VHI | HI | VHI | SMI | EI | SMI | LI | SLI | LI |
| L3 | SMI | SMI | SMI | EI | SLI | EI | VLI | LI | VLI |
| L4 | HI | HI | EI | SLI | SLI | SLI | VHI | HI | HI |
| L5 | HI | SMI | SLI | VHI | VHI | AMI | HI | SMI | SMI |
| L6 | SMI | HI | HI | VHI | HI | HI | SMI | EI | EI |
| L7 | HI | SMI | SMI | HI | SMI | SMI | SMI | SMI | EI |
| L8 | SMI | SMI | VHI | SMI | EI | EI | EI | EI | SLI |
| L9 | EI | EI | HI | SLI | SLI | SMI | SLI | SLI | LI |
| L10 | HI | SLI | SMI | LI | LI | LI | VHI | VHI | VHI |
| | | C4 | | | C5 | | | C6 | |
| L1 | HI | HI | HI | HI | HI | HI | SMI | SMI | SMI |
| L2 | HI | SMI | HI | SMI | HI | SMI | HI | HI | HI |
| L3 | SMI | EI | SMI | HI | SMI | HI | SMI | SMI | SMI |
| L4 | EI | SLI | EI | SMI | HI | SMI | HI | HI | HI |
| L5 | VHI | VHI | VHI | EI | EI | EI | VHI | VHI | VHI |
| L6 | HI | HI | HI | EI | EI | EI | SMI | SMI | SMI |
| L7 | AMI | AMI | HI | SLI | SLI | SLI | EI | EI | EI |
| L8 | VHI | VHI | VHI | SLI | LI | SLI | HI | HI | HI |
| L9 | HI | HI | HI | LI | VLI | LI | LI | LI | LI |
| L10 | HI | VHI | HI | VLI | LI | ALI | ALI | ALI | ALI |
| | | C7 | | | C8 | | | C9 | |
| L1 | VHI | VHI | VHI | VHI | VHI | VHI | EI | SLI | EI |
| L2 | SMI | EI | SMI | EI | SMI | SMI | SLI | LI | SLI |
| L3 | EI | SLI | EI | SLI | EI | EI | LI | ALI | LI |
| L4 | SLI | SLI | SLI | SLI | SLI | SLI | HI | VHI | HI |
| L5 | VHI | VHI | VHI | VHI | VHI | VHI | SMI | SMI | SMI |
| L6 | HI | HI | VHI | HI | VHI | HI | EI | EI | EI |
| L7 | SMI | SMI | HI | SMI | HI | SMI | SMI | EI | SMI |
| L8 | EI | EI | SMI | EI | SMI | EI | EI | SLI | EI |
| L9 | SLI | SLI | SLI | SLI | SLI | SLI | SLI | LI | SLI |
| L10 | SLI | LI | LI | LI | SLI | LI | VHI | VHI | VHI |

**Table 5.** *Cont.*

|  | DM1 | DM2 | DM3 | DM1 | DM2 | DM3 | DM1 | DM2 | DM3 |
|---|---|---|---|---|---|---|---|---|---|
|  |  | C10 |  |  | C11 |  |  | C12 |  |
| L1 | HI | HI | HI | EI | SLI | EI | EI | EI | SLI |
| L2 | SMI | SMI | HI | SLI | LI | SLI | SLI | SLI | LI |
| L3 | HI | HI | SMI | LI | VLI | LI | LI | LI | VLI |
| L4 | SMI | SMI | HI | HI | VHI | HI | HI | HI | VHI |
| L5 | EI | EI | EI | SMI | HI | SMI | SMI | SMI | HI |
| L6 | EI | EI | EI | EI | SMI | EI | EI | EI | SMI |
| L7 | SLI | SLI | SLI | SMI | SMI | SMI | SMI | SMI | SMI |
| L8 | SLI | SLI | LI | EI | EI | EI | EI | EI | EI |
| L9 | LI | LI | VLI | SLI | SLI | SLI | SLI | SLI | SLI |
| L10 | VLI | VLI | LI | VHI | VHI | VHI | VHI | VHI | VHI |
|  |  | C13 |  |  |  |  |  |  |  |
| L1 | EI | EI | SLI |  |  |  |  |  |  |
| L2 | SLI | SLI | LI |  |  |  |  |  |  |
| L3 | LI | LI | VLI |  |  |  |  |  |  |
| L4 | HI | HI | VHI |  |  |  |  |  |  |
| L5 | SMI | SMI | HI |  |  |  |  |  |  |
| L6 | EI | EI | SMI |  |  |  |  |  |  |
| L7 | SMI | SMI | SMI |  |  |  |  |  |  |
| L8 | EI | EI | EI |  |  |  |  |  |  |
| L9 | SLI | SLI | SLI |  |  |  |  |  |  |
| L10 | VHI | VHI | VHI |  |  |  |  |  |  |

Then, the decision matrix is transformed into a matrix of spherical fuzzy numbers by Table 1, which is presented in Table 6.

The columns of this matrix are related to the options (suggested sites for burying medical waste), and the rows represent the criteria. Then, the decision matrix formed using the SWAM operator is integrated according to the weight of the DMs shown in Table 7, and according to step (7) in the SF-WASPAS method, the values $\widetilde{Q}_i^{(1)}$, $\widetilde{Q}_i^{(2)}$, and $\widetilde{Q}_i$ are computed, and the outcomes are presented in Table 8. The ranking of the alternatives is shown with $\lambda = 0.5$.

**Table 6.** The conversion of linguistic variables related to the ranking of alternatives based on expert opinions.

|  | DM1 | DM2 | DM3 | DM1 | DM2 | DM3 |
|---|---|---|---|---|---|---|
|  |  | C1 |  |  | C2 |  |
| L1 | 0.7,0.3,0.3 | 0.7,0.3,0.3 | 0.7,0.3,0.3 | 0.7,0.3,0.3 | 0.8,0.2,0.2 | 0.8,0.2,0.2 |
| L2 | 0.8,0.2,0.2 | 0.7,0.3,0.3 | 0.8,0.2,0.2 | 0.6,0.4,0.4 | 0.5,0.5,0.5 | 0.6,0.4,0.4 |
| L3 | 0.6,0.4,0.4 | 0.6,0.4,0.4 | 0.6,0.4,0.4 | 0.5,0.5,0.5 | 0.4,0.6,0.4 | 0.5,0.5,0.5 |
| L4 | 0.7,0.3,0.3 | 0.7,0.3,0.3 | 0.5,0.5,0.5 | 0.4,0.6,0.4 | 0.4,0.6,0.4 | 0.4,0.6,0.4 |
| L5 | 0.7,0.3,0.3 | 0.6,0.4,0.4 | 0.4,0.6,0.4 | 0.8,0.2,0.2 | 0.8,0.2,0.2 | 0.9,0.1,0.1 |
| L6 | 0.6,0.4,0.4 | 0.7,0.3,0.3 | 0.7,0.3,0.3 | 0.8,0.2,0.2 | 0.7,0.3,0.3 | 0.7,0.3,0.3 |
| L7 | 0.7,0.3,0.3 | 0.6,0.4,0.4 | 0.6,0.4,0.4 | 0.7,0.3,0.3 | 0.6,0.4,0.4 | 0.6,0.4,0.4 |
| L8 | 0.6,0.4,0.4 | 0.6,0.4,0.4 | 0.8,0.2,0.2 | 0.6,0.4,0.4 | 0.5,0.5,0.5 | 0.5,0.5,0.5 |
| L9 | 0.5,0.5,0.5 | 0.5,0.5,0.5 | 0.7,0.3,0.3 | 0.4,0.6,0.4 | 0.4,0.6,0.4 | 0.6,0.4,0.4 |
| L10 | 0.7,0.3,0.3 | 0.4,0.6,0.4 | 0.6,0.4,0.4 | 0.3,0.7,0.3 | 0.3,0.7,0.3 | 0.3,0.7,0.3 |
|  |  | C3 |  |  | C4 |  |
| L1 | 0.4,0.6,0.4 | 0.5,0.5,0.5 | 0.4,0.6,0.4 | 0.7,0.3,0.3 | 0.7,0.3,0.3 | 0.7,0.3,0.3 |
| L2 | 0.3,0.7,0.3 | 0.4,0.6,0.4 | 0.3,0.7,0.3 | 0.7,0.3,0.3 | 0.6,0.4,0.4 | 0.7,0.3,0.3 |
| L3 | 0.2,0.8,0.2 | 0.3,0.7,0.3 | 0.2,0.8,0.2 | 0.6,0.4,0.4 | 0.5,0.5,0.5 | 0.6,0.4,0.4 |
| L4 | 0.8,0.2,0.2 | 0.7,0.3,0.3 | 0.7,0.3,0.3 | 0.5,0.5,0.5 | 0.4,0.6,0.4 | 0.5,0.5,0.5 |
| L5 | 0.7,0.3,0.3 | 0.6,0.4,0.4 | 0.6,0.4,0.4 | 0.8,0.2,0.2 | 0.8,0.2,0.2 | 0.8,0.2,0.2 |
| L6 | 0.6,0.4,0.4 | 0.5,0.5,0.5 | 0.5,0.5,0.5 | 0.7,0.3,0.3 | 0.7,0.3,0.3 | 0.7,0.3,0.3 |
| L7 | 0.6,0.4,0.4 | 0.6,0.4,0.4 | 0.5,0.5,0.5 | 0.9,0.1,0.1 | 0.9,0.1,0.1 | 0.7,0.3,0.3 |
| L8 | 0.5,0.5,0.5 | 0.5,0.5,0.5 | 0.4,0.6,0.4 | 0.8,0.2,0.2 | 0.8,0.2,0.2 | 0.8,0.2,0.2 |
| L9 | 0.4,0.6,0.4 | 0.4,0.6,0.4 | 0.3,0.7,0.3 | 0.7,0.3,0.3 | 0.7,0.3,0.3 | 0.7,0.3,0.3 |
| L10 | 0.8,0.2,0.2 | 0.8,0.2,0.2 | 0.8,0.2,0.2 | 0.7,0.3,0.3 | 0.8,0.2,0.2 | 0.7,0.3,0.3 |

**Table 6.** *Cont.*

|  | DM1 | DM2 | DM3 | DM1 | DM2 | DM3 |
|---|---|---|---|---|---|---|
|  | | C5 | | | C6 | |
| L1 | 0.7,0.3,0.3 | 0.7,0.3,0.3 | 0.7,0.3,0.3 | 0.6,0.4,0.4 | 0.6,0.4,0.4 | 0.6,0.4,0.4 |
| L2 | 0.6,0.4,0.4 | 0.7,0.3,0.3 | 0.6,0.4,0.4 | 0.7,0.3,0.3 | 0.7,0.3,0.3 | 0.7,0.3,0.3 |
| L3 | 0.7,0.3,0.3 | 0.6,0.4,0.4 | 0.7,0.3,0.3 | 0.6,0.4,0.4 | 0.6,0.4,0.4 | 0.6,0.4,0.4 |
| L4 | 0.6,0.4,0.4 | 0.7,0.3,0.3 | 0.6,0.4,0.4 | 0.7,0.3,0.3 | 0.7,0.3,0.3 | 0.7,0.3,0.3 |
| L5 | 0.5,0.5,0.5 | 0.5,0.5,0.5 | 0.5,0.5,0.5 | 0.8,0.2,0.2 | 0.8,0.2,0.2 | 0.8,0.2,0.2 |
| L6 | 0.5,0.5,0.5 | 0.5,0.5,0.5 | 0.5,0.5,0.5 | 0.6,0.4,0.4 | 0.6,0.4,0.4 | 0.6,0.4,0.4 |
| L7 | 0.4,0.6,0.4 | 0.4,0.6,0.4 | 0.4,0.6,0.4 | 0.5,0.5,0.5 | 0.5,0.5,0.5 | 0.5,0.5,0.5 |
| L8 | 0.4,0.6,0.4 | 0.3,0.7,0.3 | 0.4,0.6,0.4 | 0.7,0.3,0.3 | 0.7,0.3,0.3 | 0.7,0.3,0.3 |
| L9 | 0.3,0.7,0.3 | 0.2,0.8,0.2 | 0.3,0.7,0.3 | 0.3,0.7,0.3 | 0.3,0.7,0.3 | 0.3,0.7,0.3 |
| L10 | 0.2,0.8,0.2 | 0.3,0.7,0.3 | 0.1,0.9,0.1 | 0.1,0.9,0.1 | 0.1,0.9,0.1 | 0.1,0.9,0.1 |
|  | | C7 | | | C8 | |
| L1 | 0.8,0.2,0.2 | 0.8,0.2,0.2 | 0.8,0.2,0.2 | 0.8,0.2,0.2 | 0.8,0.2,0.2 | 0.8,0.2,0.2 |
| L2 | 0.6,0.4,0.4 | 0.5,0.5,0.5 | 0.6,0.4,0.4 | 0.5,0.5,0.5 | 0.6,0.4,0.4 | 0.6,0.4,0.4 |
| L3 | 0.5,0.5,0.5 | 0.4,0.6,0.4 | 0.5,0.5,0.5 | 0.4,0.6,0.4 | 0.5,0.5,0.5 | 0.5,0.5,0.5 |
| L4 | 0.4,0.6,0.4 | 0.4,0.6,0.4 | 0.4,0.6,0.4 | 0.4,0.6,0.4 | 0.4,0.6,0.4 | 0.4,0.6,0.4 |
| L5 | 0.8,0.2,0.2 | 0.8,0.2,0.2 | 0.8,0.2,0.2 | 0.8,0.2,0.2 | 0.8,0.2,0.2 | 0.8,0.2,0.2 |
| L6 | 0.7,0.3,0.3 | 0.7,0.3,0.3 | 0.8,0.2,0.2 | 0.7,0.3,0.3 | 0.8,0.2,0.2 | 0.7,0.3,0.3 |
| L7 | 0.6,0.4,0.4 | 0.6,0.4,0.4 | 0.7,0.3,0.3 | 0.6,0.4,0.4 | 0.7,0.3,0.3 | 0.6,0.4,0.4 |
| L8 | 0.5,0.5,0.5 | 0.5,0.5,0.5 | 0.6,0.4,0.4 | 0.5,0.5,0.5 | 0.6,0.4,0.4 | 0.5,0.5,0.5 |
| L9 | 0.4,0.6,0.4 | 0.4,0.6,0.4 | 0.4,0.6,0.4 | 0.4,0.6,0.4 | 0.4,0.6,0.4 | 0.4,0.6,0.4 |
| L10 | 0.4,0.6,0.4 | 0.3,0.7,0.3 | 0.3,0.7,0.3 | 0.3,0.7,0.3 | 0.4,0.6,0.4 | 0.3,0.7,0.3 |
|  | | C9 | | | C10 | |
| L1 | 0.5,0.5,0.5 | 0.4,0.6,0.4 | 0.5,0.5,0.5 | 0.7,0.3,0.3 | 0.7,0.3,0.3 | 0.7,0.3,0.3 |
| L2 | 0.4,0.6,0.4 | 0.3,0.7,0.3 | 0.4,0.6,0.4 | 0.6,0.4,0.4 | 0.6,0.4,0.4 | 0.7,0.3,0.3 |
| L3 | 0.3,0.7,0.3 | 0.1,0.9,0.1 | 0.3,0.7,0.3 | 0.7,0.3,0.3 | 0.7,0.3,0.3 | 0.6,0.4,0.4 |
| L4 | 0.7,0.3,0.3 | 0.8,0.2,0.2 | 0.7,0.3,0.3 | 0.6,0.4,0.4 | 0.6,0.4,0.4 | 0.7,0.3,0.3 |
| L5 | 0.6,0.4,0.4 | 0.6,0.4,0.4 | 0.6,0.4,0.4 | 0.5,0.5,0.5 | 0.5,0.5,0.5 | 0.5,0.5,0.5 |
| L6 | 0.5,0.5,0.5 | 0.5,0.5,0.5 | 0.5,0.5,0.5 | 0.5,0.5,0.5 | 0.5,0.5,0.5 | 0.5,0.5,0.5 |
| L7 | 0.6,0.4,0.4 | 0.5,0.5,0.5 | 0.6,0.4,0.4 | 0.4,0.6,0.4 | 0.4,0.6,0.4 | 0.4,0.6,0.4 |
| L8 | 0.5,0.5,0.5 | 0.4,0.6,0.4 | 0.5,0.5,0.5 | 0.4,0.6,0.4 | 0.4,0.6,0.4 | 0.3,0.7,0.3 |
| L9 | 0.4,0.6,0.4 | 0.3,0.7,0.3 | 0.4,0.6,0.4 | 0.3,0.7,0.3 | 0.3,0.7,0.3 | 0.2,0.8,0.2 |
| L10 | 0.8,0.2,0.2 | 0.8,0.2,0.2 | 0.8,0.2,0.2 | 0.2,0.8,0.2 | 0.2,0.8,0.2 | 0.3,0.7,0.3 |
|  | | C11 | | | C12 | |
| L1 | 0.5,0.5,0.5 | 0.4,0.6,0.4 | 0.5,0.5,0.5 | 0.5,0.5,0.5 | 0.5,0.5,0.5 | 0.4,0.6,0.4 |
| L2 | 0.4,0.6,0.4 | 0.3,0.7,0.3 | 0.4,0.6,0.4 | 0.4,0.6,0.4 | 0.4,0.6,0.4 | 0.3,0.7,0.3 |
| L3 | 0.3,0.7,0.3 | 0.2,0.8,0.2 | 0.3,0.7,0.3 | 0.3,0.7,0.3 | 0.3,0.7,0.3 | 0.2,0.8,0.2 |
| L4 | 0.7,0.3,0.3 | 0.8,0.2,0.2 | 0.7,0.3,0.3 | 0.7,0.3,0.3 | 0.7,0.3,0.3 | 0.8,0.2,0.2 |
| L5 | 0.6,0.4,0.4 | 0.7,0.3,0.3 | 0.6,0.4,0.4 | 0.6,0.4,0.4 | 0.6,0.4,0.4 | 0.7,0.3,0.3 |
| L6 | 0.5,0.5,0.5 | 0.6,0.4,0.4 | 0.5,0.5,0.5 | 0.5,0.5,0.5 | 0.5,0.5,0.5 | 0.6,0.4,0.4 |
| L7 | 0.6,0.4,0.4 | 0.6,0.4,0.4 | 0.6,0.4,0.4 | 0.6,0.4,0.4 | 0.6,0.4,0.4 | 0.6,0.4,0.4 |
| L8 | 0.5,0.5,0.5 | 0.5,0.5,0.5 | 0.5,0.5,0.5 | 0.5,0.5,0.5 | 0.5,0.5,0.5 | 0.5,0.5,0.5 |
| L9 | 0.4,0.6,0.4 | 0.4,0.6,0.4 | 0.4,0.6,0.4 | 0.4,0.6,0.4 | 0.4,0.6,0.4 | 0.4,0.6,0.4 |
| L10 | 0.8,0.2,0.2 | 0.8,0.2,0.2 | 0.8,0.2,0.2 | 0.8,0.2,0.2 | 0.8,0.2,0.2 | 0.8,0.2,0.2 |
|  | | C13 | | | | |
| L1 | 0.5,0.5,0.5 | 0.5,0.5,0.5 | 0.4,0.6,0.4 | | | |
| L2 | 0.4,0.6,0.4 | 0.4,0.6,0.4 | 0.3,0.7,0.3 | | | |
| L3 | 0.3,0.7,0.3 | 0.3,0.7,0.3 | 0.2,0.8,0.2 | | | |
| L4 | 0.7,0.3,0.3 | 0.7,0.3,0.3 | 0.8,0.2,0.2 | | | |
| L5 | 0.6,0.4,0.4 | 0.6,0.4,0.4 | 0.7,0.3,0.3 | | | |
| L6 | 0.5,0.5,0.5 | 0.5,0.5,0.5 | 0.6,0.4,0.4 | | | |
| L7 | 0.6,0.4,0.4 | 0.6,0.4,0.4 | 0.6,0.4,0.4 | | | |
| L8 | 0.5,0.5,0.5 | 0.5,0.5,0.5 | 0.5,0.5,0.5 | | | |
| L9 | 0.4,0.6,0.4 | 0.4,0.6,0.4 | 0.4,0.6,0.4 | | | |
| L10 | 0.8,0.2,0.2 | 0.8,0.2,0.2 | 0.8,0.2,0.2 | | | |

**Table 7.** The aggregated decision matrix based on SWAM operator.

|  | **C1** | **C2** | **C3** | **C4** | **C5** |
|---|---|---|---|---|---|
| L1 | 0.7,0.3,0.3 | 0.77,0.23,0.23 | 0.42,0.58,0.43 | 0.70,0.30,0.30 | 0.70,0.30,0.70 |
| L2 | 0.78,0.22,0.22 | 0.58,0.42,0.42 | 0.32,0.68,0.33 | 0.68,0.32,0.32 | 0.62,0.38,0.38 |
| L3 | 0.6,0.4,0.4 | 0.48,0.52,0.49 | 0.22,0.78,0.23 | 0.58,0.42,0.42 | 0.68,0.32,0.32 |
| L4 | 0.62,0.39,0.40 | 0.40,0.60,0.40 | 0.74,0.27,0.27 | 0.48,0.52,0.49 | 0.62,0.38,0.38 |
| L5 | 0.56,0.45,0.37 | 0.86,0.14,0.15 | 0.63,0.37,0.37 | 0.80,0.20,0.20 | 0.50,0.50,0.50 |
| L6 | 0.67,0.33,0.33 | 0.74,0.27,0.27 | 0.53,0.47,0.47 | 0.70,0.30,0.30 | 0.50,0.50,0.50 |
| L7 | 0.63,0.37,0.37 | 0.63,0.37,0.37 | 0.55,0.45,0.45 | 0.83,0.17,0.19 | 0.40,0.60,0.40 |
| L8 | 0.72,0.28,0.30 | 0.53,0.47,0.47 | 0.45,0.55,0.46 | 0.80,0.20,0.20 | 0.38,0.62,0.38 |
| L9 | 0.62,0.39,0.40 | 0.51,0.49,0.40 | 0.35,0.65,0.36 | 0.70,0.30,0.30 | 0.28,0.72,0.28 |
| L10 | 0.61,0.40,0.37 | 0.30,0.70,0.30 | 0.80,0.20,0.20 | 0.72,0.28,0.28 | 0.19,0.83,0.19 |
|  | **C6** | **C7** | **C8** | **C9** | **C10** |
| L1 | 0.60,0.40,0.40 | 0.80,0.20,0.20 | 0.80,0.20,0.20 | 0.48,0.52,0.49 | 0.70,0.30,0.30 |
| L2 | 0.70,0.30,0.30 | 0.58,0.42,0.42 | 0.5,0.43,0.43 | 0.38,0.62,0.38 | 0.65,0.35,0.35 |
| L3 | 0.60,0.40,0.40 | 0.48,0.52,0.49 | 0.47,0.53,0.48 | 0.27,0.74,0.28 | 0.65,0.35,0.35 |
| L4 | 0.70,0.30,0.30 | 0.40,0.60,0.40 | 0.40,0.60,0.40 | 0.72,0.28,0.28 | 0.65,0.35,0.35 |
| L5 | 0.80,0.20,0.20 | 0.80,0.20,0.20 | 0.80,0.20,0.20 | 0.60,0.40,0.40 | 0.50,0.50,0.50 |
| L6 | 0.60,0.40,0.40 | 0.76,0.24,0.25 | 0.72,0.28,0.28 | 0.50,0.50,0.50 | 0.50,0.50,0.50 |
| L7 | 0.50,0.50,0.50 | 0.65,0.35,0.35 | 0.62,0.38,0.38 | 0.58,0.42,0.42 | 0.40,0.60,0.40 |
| L8 | 0.70,0.30,0.30 | 0.55,0.45,0.45 | 0.52,0.48,0.48 | 0.48,0.52,0.49 | 0.35,0.65,0.36 |
| L9 | 0.30,0.70,0.30 | 0.40,0.60,0.40 | 0.40,0.60,0.40 | 0.38,0.62,0.38 | 0.26,0.75,0.26 |
| L10 | 0.10,0.90,0.10 | 0.33,0.67,0.34 | 0.32,0.68,0.33 | 0.80,0.20,0.20 | 0.26,0.75,0.26 |
|  | **C11** | **C12** | **C13** |  |  |
| L1 | 0.48,0.52,0.55 | 0.45,0.55,0.46 | 0.45,0.55,0.46 |  |  |
| L2 | 0.38,0.62,0.44 | 0.35,0.65,0.36 | 0.35,0.65,0.36 |  |  |
| L3 | 0.28,0.72,0.33 | 0.26,0.75,0.26 | 0.26,0.75,0.26 |  |  |
| L4 | 0.72,0.28,0.43 | 0.76,0.24,0.25 | 0.76,0.24,0.25 |  |  |
| L5 | 0.62,0.38,0.48 | 0.65,0.35,0.35 | 0.65,0.35,0.35 |  |  |
| L6 | 0.52,0.48,0.54 | 0.55,0.45,0.45 | 0.55,0.45,0.45 |  |  |
| L7 | 0.60,0.40,0.50 | 0.60,0.40,0.40 | 0.60,0.40,0.40 |  |  |
| L8 | 0.50,0.50,0.56 | 0.50,0.50,0.50 | 0.50,0.50,0.50 |  |  |
| L9 | 0.40,0.60,0.45 | 0.40,0.60,0.40 | 0.40,0.60,0.40 |  |  |
| L10 | 0.80,0.20,0.41 | 0.80,0.20,0.20 | 0.80,0.20,0.20 |  |  |

**Table 8.** The spherical fuzzy values of $\widetilde{Q}_i^{(1)}$, $\widetilde{Q}_i^{(2)}$, and $\widetilde{Q}_i$.

|  | $\widetilde{Q}_i^{(1)}$ **(WSM)** | $\widetilde{Q}_i^{(2)}$ **(WPM)** | $\widetilde{Q}_i$ | **Normalized Def. Values** | **Rank** |
|---|---|---|---|---|---|
| L1 | 0.65,0.35,0.33 | 0.60,0.41,0.38 | 0.63,0.38,0.36 | 0.071497 | 2 |
| L2 | 0.58,0.42,0.34 | 0.52,0.49,0.36 | 0.55,0.45,0.35 | 0.0303885 | 7 |
| L3 | 0.49,0.52,0.38 | 0.42,0.59,0.36 | 0.45,0.55,0.37 | −0.027326 | 9 |
| L4 | 0.64,0.36,0.33 | 0.59,0.42,0.36 | 0.61,0.39,0.35 | 0.069506 | 3 |
| L5 | 0.71,0.29,0.29 | 0.67,0.34,0.34 | 0.69,0.31,0.32 | 0.1380487 | 1 |
| L6 | 0.62,0.37,0.38 | 0.60,0.40,0.41 | 0.61,0.39,0.40 | 0.0459783 | 5 |
| L7 | 0.62,0.38,0.37 | 0.58,0.42,0.39 | 0.60,0.40,0.38 | 0.0462505 | 4 |
| L8 | 0.59,0.41,0.40 | 0.54,0.46,0.43 | 0.56,0.44,0.41 | 0.0227096 | 8 |
| L9 | 0.46,0.54,0.37 | 0.41,0.59,0.36 | 0.44,0.57,0.36 | −0.035712 | 10 |
| L10 | 0.64,0.38,0.25 | 0.45,0.59,0.25 | 0.56,0.48,0.26 | 0.0421991 | 6 |

## 6. Comparison and Sensitivity Analysis

In this section, in order to show the reliability and accuracy of output obtained and the ability of the suggested approach, the prioritization of proposed landfills for medical waste is compared with IF-SWARA-WASPAS. According to Table 9, it can be seen that according to the SF-WASPAS method, L5, L1, and L4, with scores 0.1380487, 0.071497 and 0.069506, are in the first to third priorities, respectively. With a general review of the outcomes, we find that in the IF-WASPAS method, L5 and L7 are jointly in the first place with equal points, and this indicates that the prioritization based on the intuitive fuzzy method has not been performed completely and the resolution is not possible, and the decision-maker

may be confused and mistaken. Comparing the intuitive and spherical fuzzy results, it can be seen that the priorities have changed. This shows that SFS, while retaining the advantages of the IF-WASPAS method, can better consider uncertainty and has a greater degree of freedom to express specialists' opinions about the environment.

**Table 9.** The comparison of the proposed approach results with IF area.

| ALTERNATIVE | SF-SWARA-WASPAS | | IF-SWARA-WASPAS | |
|---|---|---|---|---|
| | SCORE | RANK | SCORE | RANK |
| L1 | 0.071497 | 2 | 0.719 | 2 |
| L2 | 0.030388 | 7 | 0.626 | 7 |
| L3 | −0.02733 | 9 | 0.5 | 8 |
| L4 | 0.069506 | 3 | 0.698 | 3 |
| L5 | 0.138049 | 1 | 0.999 | 1 |
| L6 | 0.045978 | 5 | 0.697 | 4 |
| L7 | 0.04625 | 4 | 0.999 | 1 |
| L8 | 0.02271 | 8 | 0.6645 | 6 |
| L9 | −0.03571 | 10 | 0.4907 | 9 |
| L10 | 0.042199 | 6 | 0.6721 | 5 |

In this section, to perform sensitivity analysis on the criteria of the proposed method, ten different scenarios were created by changing the weight of the criteria, and the results were compared. According to Figure 3, L5 and L9 are suitable and unsuitable landfills in all ten scenarios, respectively, and although the ranking results have changed in a number of scenarios, the overall results are similar to the SF-SWARA-WASPAS method.

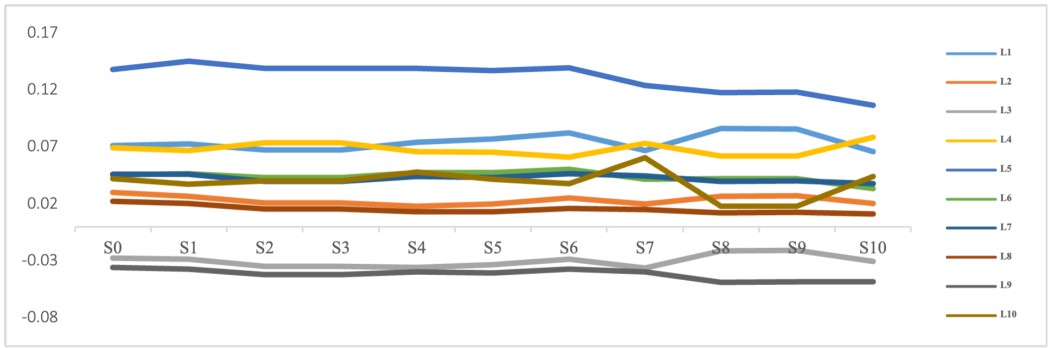

**Figure 3.** Comparison of different scenarios.

## 7. Conclusions

### 7.1. Discussion on Managerial Implications

Waste control, including toxic and hazardous waste, part of which is hospital waste, is inevitable in municipal waste management. Every day, the existence of thousands of tons of sanitary waste in different cities of the country is an issue that should be at the top of the country's health and environmental programs due to the increase in population and the development of industry and technology. Furthermore, the waste of health centers due to pollution is a reservoir of pathogenic microorganisms that can cause infection, as well as increase a variety of dangerous and contagious diseases and pose a serious threat to human health and the environment. In recent years, the importance of waste disposal to control its adverse effects on the environment has increased, and among waste disposal methods, landfilling is one of the best methods, and in this regard, choosing a suitable place for burying medical waste is an important process. It should be performed with the care and cooperation of departments and institutions, such as environmental protection, environmental health, regional water organization, agriculture, and natural resources. Important factors in choosing a landfill should be considered. In this study, we tried to pay attention to the importance of the issue and the basic factors with regard to the general



health of human beings, and the most appropriate place for landfilling of health waste was selected. Therefore, a model has been utilized that considers the degree of membership criteria, the degree of non-membership, and its doubts in the environment of uncertainty. If applied, its reflection will have a major impact on maintaining the health of the community environment.

### 7.2. Conclusions

This study presents an integrated SWARA-WASPAS approach based on a spherical fuzzy environment for selecting a medical waste landfill site with the purpose of overcoming the shortcomings of traditional MCDM methods. The present study presents arithmetic operations with spherical fuzzy numbers and proposes a numerical example to show its efficiency. According to the field and geographical situations of the region and former polls and specialist opinions, selected criteria were specified, and by using the SF-SWARA method, criteria weight was computed. Ten potential locations for the landfilling of medical wastes depending on the geographical site were presented and considered near Urmia, and by using the SFWASPAS method, these sites were ranked, and L5 was obtained as the optimal suitable site for burying medical waste. In this study, the spherical fuzzy theory was used to better express the intent of specialists and better consider uncertainty, and this environment allows decision-makers to comment more freely, and therefore, the outcomes are close to reality and can be a good guide for managers related to waste and environmental organizations. After analyzing the results, the proposed approach was compared with IF-SWARA-WASPAS, and the outcomes showed that the prioritization of this method has better considered the uncertainty related to the comments, has proposed a complete and clear prioritization, and the decision-maker can choose a suitable site for burying medical waste. SFSWARA and SFWASPAS methods give specialists the opportunity to reflect their opinions more effectually and realistically because actual information from the medical waste landfill is somewhat uncertain. This method can be used in different areas of decision-making, risk assessment, and information management. In future research, causal relationships between criteria can be considered, and novel criteria can be added according to the nature of the issue.

**Author Contributions:** S.J.G.: conceptualization, investigation, validation, and writing—review and editing; S.R.B.: writing—original draft, methodology, software, and investigation; A.M.G.: writing—original draft, conceptualization, investigation, and methodology; G.H.: data curation, formal analysis, and writing—review and editing; H.T.: validation, investigation, and visualization; M.H.-K.: writing—review and editing, validation, and conceptualization. All authors have read and agreed to the published version of the manuscript.

**Funding:** The APC was funded by FIM UHK Excellence Project 2021: Decision Support Systems: Principles and Applications 3.

**Institutional Review Board Statement:** Not applicable.

**Informed Consent Statement:** Not applicable.

**Data Availability Statement:**

**Conflicts of Interest:** The authors declare no conflict of interest.

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
