# Peer review of "Landfill Site Selection for Medical Waste Using an Integrated SWARA-WASPAS Framework Based on Spherical Fuzzy Set"

_sustainability, doi:10.3390/su132413950_

Round 1

Reviewer 2 Report

Dear Authors

The presented study on “Landfill site selection for medical waste using an integrated SWARA-WASPAS framework based on Spherical Fuzzy Set” design and written very well by the authors. The data are also statically presented in well manner.  The manuscript could be potential in the management of medical waste i.e. general, infectious, hazardous and radioactive etc. to reducing the risk of health and environment. Therefore, I strongly recommended minor corrections (Below).

    1. The language of abstract section needs improvement
    2. Provide information about important data / outcomes in abstract section
    3. Line 43: PPEs? Do not use abbreviation first time
    4. Line 67-68 Write full form of TOPSIS, SWARA, MOORA, WAPAS

Round 2

Reviewer 1 Report

The authors did a great job in revising the manuscript. Hence, I recommend for publication in this current form.